# Secretome Analysis of High- and Low-Virulent Bovine *Pasteurella multocida* Cultured in Different Media

**DOI:** 10.3390/ani13233683

**Published:** 2023-11-28

**Authors:** Yangyang Qiu, Jianan Wang, Fang He, Xiaoyan Wu, Ruitong Dan, Philip R. Hardwidge, Nengzhang Li, Yuanyi Peng

**Affiliations:** 1College of Veterinary Medicine, Southwest University, Chongqing 400715, China; wangjianan0826@163.com (J.W.); hefang2017@sina.com (F.H.); xlhylykxmb@163.com (R.D.); lich2001020@163.com (N.L.); 2College of Animal Science, South China Agricultural University, Guangzhou 510642, China; xiaoyanwu2020@126.com; 3College of Veterinary Medicine, Kansas State University, Manhattan, KS 66502, USA; hardwidg@vet.k-state.edu

**Keywords:** *Pasteurella multocida*, proteomics, (DIA) LC-MS/MS, secreted protein, subunit vaccine

## Abstract

**Simple Summary:**

The absence of secretome information of *P. multocida* impedes the understanding of the mechanisms of infection and the development of subunit vaccines of *P. multocida.* In this study, we employed data-independent acquisition (DIA) LC-MS/MS combined with bioinformatics analysis to obtain more comprehensive and accurate information on proteins secreted by *P. multocida.* Then, the selected Tuf protein was verified by bioinformatics analysis, immunohistochemistry and immunoelectron microscopy. In addition, in this study, Tuf protein was expressed as recombinant antigen and tested as a subunit vaccine, and its biological characteristics were studied by serum antibody titer detection and *Pasteurella* survival assay. As a result, this study aims to offer a novel perspective and solid theoretical foundation for the investigation of the pathogenic mechanism of *P. multocida* and the design of vaccines.

**Abstract:**

Bovine *Pasteurella multocida* (*P. multocida*) serotype A is one of the major causes of bovine respiratory disease (BRD). We used data-independent acquisition (DIA) LC-MS/MS combined with bioinformatics analysis to identify proteins secreted by *P. multocida*. A total of 154 proteins were obtained from the supernatants of two isolates of bovine *P. multocida* serotype A (high virulent PmCQ2 and low virulent PmCQ6) cultured in Martin or BHI media, of which 50 were identified as putative secreted proteins. Further studies showed that Tuf, an elongation factor Tu, was highly expressed in *P. multocida* and secreted into infected tissues. Tuf stimulated strong innate immune responses of macrophages and had protective efficacy against *P. multocida* infection in a mouse model. The results provide insight into the secreted proteins of *P. multocida* and suggest new targets for vaccine development against *P. multocida*.

## 1. Introduction

*Pasteurella multocida (P. multocida*) is an important bacterial pathogen that causes bovine respiratory disease complex (BRDC) and hemorrhagic septicemia. *P. multocida* can be divided into five serotypes: A, B, D, E, and F [1]. BRDC caused by *P. multocida* serotype A is currently more common and severe than other serotypes. Currently, the abuse of antibiotics has led to the increased drug resistance of *P. multocida* [2]; therefore, there is an urgent need to develop vaccines against *P. multocida* infection. At present, many virulence factors (VFs) in *P. multocida* including lipopolysaccharides (LPS), outer-membrane proteins (OMP), *P. multocida* toxins (PMT), iron-regulated outer-membrane proteins (IROMP), and capsules (CP) are well understood. However, the absence of secretome information of *P. multocida* still impedes the understanding of the mechanisms of infection and the development of subunit vaccines of *P. multocida*.

PmCQ2 (high virulence) and PmCQ6 (low virulence) serotype A, previously isolated from pneumonic lungs, share 99% genomic similarity [3]. Moreover, previous work has shown that the low virulence PmCQ6 strain has a mutation in the start codon of hyaC and thus reduced levels of capsule, a known crucial virulence factor [4]. Nonetheless, transcriptome sequencing analysis of PmCQ2 and PmCQ6 revealed abundant DEGs (differentially expressed genes) [5]. To investigate whether these DEGs are also associated with the virulence of *P. multocida*, we performed proteomic sequencing to find critical functional proteins, and wanted to provide a theoretical basis for the pathogenesis of *P. multocida* and vaccine development. In the process of proteomic analysis, we focus on predicted secreted proteins. Several effector proteins encoded by bacteria are secreted extracellularly through their nanomachines (such as type III, type IV, and type VI secretion systems) [6]. Studies have found that one-third of the proteins synthesized in Gram-negative bacteria need to be transported or secreted for extracellular function, including some hydrolases, cytotoxins, growth factors, hormones, or antibodies [7,8].

According to the presence or absence of signal peptide, the bacterial protein secretory pathways are roughly divided into two types. One includes signal peptide-dependent pathways, such as Sec secretion pathway, Tat secretion pathway, SRP (signal recognition particle) pathway, ABC (ATP-binding cassette transporter) transporter pathway, type III secretion system (T3SS), type IV secretion system (T4SS), and type VII secretion system (T7SS) [9]; the other is a non-classical secretory pathway that does not require signal peptides such as outer-membrane vesicle (OMV) transporter [10]. We can predict and classify secreted proteins by identifying how proteins are secreted according to the properties of the protein detection in bacterial culture supernatant by LC-MS/MS. Combined with bioinformatics, this can assist in the identification of the secretome, the bioinformatics secretion signal. At present, tools primarily use the presence or absence of signal peptides, as well as the location (N or C terminal) and amino acid composition of signal peptides to predict whether and how bacteria secrete proteins. To obtain more accurate information, we combined various prediction tools (SignalP, TatP, SecretomeP, LipoP, Tmhmm, and Phobius) to determine the information of secreted proteins, which is conducive to the in-depth verification of our subsequent experiments.

In this study, the culture supernatant proteins of two bovine *P. multocida* serotype A strains (PmCQ2 and PmCQ6) cultured with Martin broth and brain heart infusion broth (BHI) were detected by LC-MS/MS. The secretory proteins of both strains were analyzed by bioinformatic methods, and one secreted protein was identified based on immuno-electron microscopy. The aim of this study was to explore the potential virulence factors and protective antigens in secreted proteins, so as to provide a theoretical basis for vaccine development.

## 2. Materials and Methods

### 2.1. Bacterial Strains and Cultural Conditions

Both *P. multocida* capsular serotype A strains PmCQ2 (GeneBank accession No. CP033599) and PmCQ6 (GeneBank accession No. CP033600) were isolated from the lungs of dead calves that suffered from pneumonia in Chongqing, China [3]. The virulence of PmCQ2 (intramuscular route: LD_50_ = 3.4 × 10^3^ CFU) was higher than PmCQ6 (intramuscular route: LD_50_ = 1.14 × 10^8^ CFU) [3]. Three to four colonies of each strain were picked from Martin broth agar plate, inoculated in 5 mL Martin broth, and incubated overnight at 37 °C with shaking at 220 r/min.

### 2.2. Experimental Animals and Ethics Statement

All mouse use was approved by the experimental animal ethics committee of Southwest University (Chongqing, China) (Permit No.: IACUC-20201020-02). Kunming mice (female, 7–8-week-old) were purchased from the Hunan SJA Laboratory Animal Co., Ltd. (Changsha, China). A total of 50 Kunming mice were used in this study. Mice were raised in a special mouse culture system that was individually ventilated and free of pathogens (temperature at 20–30 °C, relative humidity at 50–60%, and lighting cycle at 12 h/day) with free food and water.

### 2.3. Growth Curve Analysis

Overnight cultures of PmCQ2 and PmCQ6 were inoculated 1:100 (*v*/*v*) into 5 mL fresh Martin broth medium supplemented with 5% horse serum, respectively, and incubated at 37 °C with shaking at 220 r/min until the OD_600_ values were about 1.0. Thereafter, they were inoculated at 1:50 (*v*/*v*) into 50 mL fresh Martin and BHI broth, respectively, and then incubated at 37 °C without shaking. The absorbance of Martin and BHI broth culture groups were measured every 3 h. The results were statistically analyzed by *t*-test.

### 2.4. Virulence Analysis of Bacteria Cultured in Different Media

Kunming mice were randomly divided into 5 groups (*n* = 10/group); 4 groups of mice were intramuscularly infected with PmCQ2 (1 × 10^7^ CFU) and PmCQ6 (5 × 10^7^ CFU), respectively, and the control group received an equal amount of normal saline. Mice were monitored for 7 days, and mice with serious clinical signs (low energy, slow reaction, no feed and water intake, eyes closed, and faint breathing) were considered moribund, and were humanely euthanized by intraperitoneal injection of 100 μL (1.5%) of sodium pentobarbital. The number of mortalities was recorded daily.

### 2.5. Sample Preparation

Overnight cultures (refer to Section 2.1 for specific culture methods) of PmCQ2 and PmCQ6 were inoculated 1:50 (*v*/*v*) into 50 mL Martin and BHI broth, respectively, and incubated at 37 °C without shaking (static culture is conducive to protein expression). The cultures were harvested when the OD_600_ reached 0.5–0.6. Each culture had 3 biological replicates and a total of 12 cultures were prepared for secretome profiles. The proteins in the culture supernatant were prepared by using the TCA protein Precipitation Protocol, and protein precipitates were quick frozen and kept in dry ice, and sent to Applied Protein Technology Company (Shanghai, China) for proteome sequencing analysis.

### 2.6. LC-MS/MS and DIA Quantitative Analysis

All protein samples were mixed for enzyme digestion and HPRP (high range resolution profile) classification. After data-dependent acquisition (DDA) mass spectrometry, MaxQuant (Maxquant_1.5.3.17) was used for database (Human UniProt) search, and the identified proteins must meet the set filtration parameters FDR < 1%. The Spectronaut (Spectronaut Pulsar X_12.0.20491.4) was used to build a spectral library for subsequent data-independent acquisition (DIA) [11]. DIA analysis was performed for each sample after enzymatic hydrolysis. The DIA raw files (FDR < 1%) were imported into the same Spectronaut for analysis, then qualitative and quantitative protein data of the samples were obtained from Applied Protein Technology Company.

### 2.7. Bioinformatics Analysis

To further identify the secreted proteins in the sequenced data, several corresponding analysis online platforms were employed. First, SignalP (http://www.cbs.dtu.dk/services/SignalP-5.0/, accessed on 3 December 2020) was used to screen out proteins with signal peptides [12], and then TMHMM v2.0 (http://www.cbs.dtu.dk/services/TMHMM/, accessed on 3 December 2020) was used to select proteins with 0 or 1 transmembrane helix. Phobius (http://phobius.sbc.su.se/, accessed on 3 December 2020) was used to exclude proteins with transmembrane regions. The above screened proteins were verified in LipoP (http://www.cbs.dtu.dk/services/LipoP/, accessed on 3 December 2020) to determine the type of signal peptidase. In summary, a protein with a signal peptide, no transmembrane helix, and recognized by type I or type II signal peptidases could be preliminarily predicted as a Sec secreted protein.

TatP (http://www.cbs.dtu.dk/services/TatP/, accessed on 3 December 2020) was used to screen out proteins with RR signal peptides and D values greater than the default threshold of 0.36, then TMHMMV 2.0 and Phobius were used to select proteins with no transmembrane helix. Finally, using SecretomeP 2.0 (http://www.cbs.dtu.dk/services/SecretomeP/, accessed on 3 December 2020), the proteins with signal peptide were removed to select the sequence with secP value greater than 0.5. A moonlight protein blast search was carried out using the following website (http://www.moonlightingproteins.org, accessed on 3 December 2020).

### 2.8. Protein Expression and Polyclonal Antibody Preparation

Based on bioinformatics prediction and secretion level of the protein, Tuf, one of the proteins with high secretion in both strains cultured in both media, was selected to further verify its secretion and localization in infection tissue and immune protection. The specific primer of rTuf (5′-3′: GGCCATGGCTGATATCGGATCCATGTCTAAAGAA-AAATTTGAACGTAC, 3′-5′: CTCGAGTGCGGCCGCAAGCTTGATAATTTTCG-CTACAACACCC) was designed using BioXM 2.6 software. The primers were synthesized by Sangon Biotech (Shanghai, China). The DNA gene fragment *tuf* was inserted into pET-30a vector (Novagen, Darmstadt, Germany) between *Bam*HI and *Xho*I. Then, the recombinant plasmid was transformed into *E. coli* strain BL21 (GenStar, Beijing, China) and treated with isopropyl β-D-1-thiogalactoside (IPTG) (0.4 mM) to induce rTuf expression. Finally, rTuf was purified using Ni-affinity chromatography and desalinated using ZebaTMSpin Desalting Columns, 7K MWCO (Thermo, Waltham, MA, USA). Endotoxin was removed by ToxinEraser^TM^ Endotoxin Removal Kit (Genscript, Piscataway, NJ, USA) and detected by a ToxinSensor TM Chromogenic LAL Endotoxin Assay Kit (Genscript, Piscataway, NJ, USA). The purified rTuf was mixed with Freund’s complete adjuvant (Sigma, St. Louis, Mo, USA) at a ratio of 4:1. After full emulsification with an oscillator, mice were immunized (100 µg/mice) via subcutaneous route with a booster immunization at 7 days after primary immunization. After one week of booster immunization, the mice were euthanized, then the serum was collected and stored at −20 °C.

### 2.9. Immunohistochemistry

The gradient diluted culture of PmCQ2, cultured to the logarithmic stage, was intraperitoneally injected into mice (1 × 10^7^ CFU/mouse). Mice with apparent pathological changes were euthanized, and their lungs were obtained and fixed with paraformaldehyde. IHC (immunohistochemistry) experiments were performed as documented in previous studies [13]. The fixed tissues were dehydrated using an automatic dehydrator, embedded, sectioned, and mounted on microscopic slides and soaked in 3% methanol hydrogen peroxide at room temperature for 10 min, and then washed 3 times with PBS. Secondly, the slides were immersed in 0.01 M citric acid buffer solution (pH 6.0), heated in the microwave oven at medium and high power for an interval of 5 min until boiling, and then the power was turned off; this step was repeated once. After cooling, the slides were washed twice with PBS for 5 min each time. Then, the slides were treated with goat serum blocking solution at room temperature for 20 min. After that, primary antibody was added and treated overnight at 4 °C. Next, biotin secondary antibody was added and incubated at 37 °C for 30 min. Thereafter, the slides were washed 3 times with PBS. Finally, DAB (diaminobenzidine) color kit was used for color development. The reagent was mixed and dropped on the slides, treated for about 2 min, and washed 2 times with distilled water. After gentle restaining with hematoxylin and dehydration, transparent and neutral gum were used to seal the slides, and six images were collected from each slide.

### 2.10. Immunoelectron Microscopy Observation

Firstly, a small piece of mouse lung tissue was taken according to the above immunohistochemical method, and immediately placed into 3% glutaraldehyde fixative solution for fixation. After fixation, it was progressively dehydrated with acetone, resin embedded, and polymerized. Then, the ultrathin sections of 60~80 nm were made and placed on a 300 mesh nickel grid. Next, a layer of Parafilm was pasted on the dyeing plate in the dyeing wet box, and 3% hydrogen peroxide liquid was dropped on the Parafilm to form water droplets. The nickel grid sections were etched upside down on the water droplets for 30 min and then washed. After washing, 1% calf serum was added, and after 20 min at room temperature, the calf serum was absorbed, and the first antibody (primary antibody prepared by immunizing mice with rTuf) was added and incubated at 4 °C for 20 h before washing. After washing, 1% calf serum was added and treated at room temperature for 20 min. Then, the calf serum was removed and colloidal gold labeled second antibody was added at room temperature for 2 h. Nickel grids were washed 5 times with PBS. Secondly, uranium acetate and lead citrate were negatively stained, then double distilled water was used 5 times to wash the grids. Finally, after its natural drying, the image of the copper net was collected using JEM-1400 Flash transmission electron microscope produced by Japan Electronics.

### 2.11. Immune Protection Assay

Overall, twenty female Kunming mice were randomly divided into 2 groups (*n* = 10/group). The purified rTuf was mixed with Freund’s complete adjuvant (Sigma, USA) in a ratio of 4:1; each mouse was immunized with 100 μg recombinant protein. The control group was immunized with the same amount of normal saline mixed with adjuvant. After the adjuvant and proteins were fully emulsified with an oscillator, the mixture was injected into mice via subcutaneous route with one booster immunization 7 days after the first immunization, then 100 μL tail vein bloods were collected at day 21 for the serum separation. After 14 days of the second immunization, the mice were challenged with PmCQ2 (1 × 10^7^ CFU) via intramuscular route. The death of mice was recorded and observed every 12 h for a week, and the survival curve of mice was drawn.

### 2.12. Serum Antibody Titer Detemination

rTuf was diluted to 1 μg/100 μL with a coating solution, then added to the HRP (horseradish peroxidase)-labeled well at the rate of 100 μL/well, and coated overnight at 4 °C. Thereafter, the washing solution (0.05% PBST buffer) was added to 300 μL/well, left to stand for 5 min, and the liquid in the well was shaken off. This procedure was repeated 5 times. Next, the blocking solution was added to the enzyme-labeled well at the rate of 200 μL/well, sealed at 37 °C for 1 h, then washed as mentioned above. The serum collected earlier was used as the primary antibody, and the serum was diluted at a certain dilution ratio, then added to the enzyme-labeled well at the rate of 100 μL/well, incubated at 37 °C for 1 h, and washed as mentioned above. Thereafter, goat anti-mouse IgG was diluted according to the instructions, and added to the enzyme-labeled well at the rate of 100 μL/well, then incubated at 37 °C for 1 h and washed as mentioned above. Finally, TMB substrate (100 μL/well) was added to each well, treated for 10 min without light, and then the stop solution (100 μL/well) was added to each well to stop the reaction. The OD_450_ was detected by ELISA reader.

### 2.13. rTuf Induces Inflammatory Factor Secretion in Macrophage

Different concentrations of soluble protein rTuf were co-incubated with mouse macrophage RAW264.7 (Thermo, Germany) for 24 h. Briefly, the adherent cells were cultured in 6-well microplates (1 × 10^6^ cells/well) with RPMI 1640 medium and different concentrations of soluble protein rTuf were added for 24 h. At the same time, LPS, boiled rTuf, and protease K + rTuf were set to exclude the interference of other factors, and ELISA kit (Thermo, Germany) was used to detect the secretion of related inflammatory factors.

### 2.14. Statistical Analysis

Data were expressed as mean ± mean standard error (SEM). The student t-test was used for a single comparison with GraphPad Prism Version 6.0 software (GraphPad Software, La Jolla, CA, USA) if the data were in Gaussian distribution and had equal variance, or by the unpaired t-test with Welch’s correction (Prism 6.0) if the data were in Gaussian distribution but showed unequal variance, or by the non-parametric test (Mann-Whitney U test, Prism 6.0) if the data were not normally distributed. The Gaussian distribution of data was analyzed by D’Agootino-Pearson omnibus normality test (Prism 6.0) and Kolmogorov-Smirnov test (Prism 6.0). Data variance was analyzed by Brown-Forsyth test (Prism 6.0). *p* < 0.05 was considered significant.

## 3. Results

### 3.1. The Virulence and Growth Level of PmCQ2 and PmCQ6 Were Different in Different Media

To explore the role of secreted proteins in the virulence differences between PmCQ2 and PmCQ6, we first predicted putative secreted proteins by analyzing genomic data of *P. multocida*. Of the 414 hypothetical secreted proteins predicted based on *P. multocida* genomic data [5], 205 proteins (about 49%) were secreted through the Sec pathway, of which 137 proteins were cleaved by SP I (33%) and 68 proteins were cleaved by SP II (16%). The proportion of Tat pathway secretory proteins was 6%, only 25 proteins. There were 186 proteins secreted through the non-classical secretion pathways, accounting for 45% (Appendix A).

Considering the differences in virulence and growth level of PmCQ2 and PmCQ6 in Martin and BHI broth (Appendix A), proteomic sequencing of PmCQ2 and PmCQ6 in different media was carried out concurrently. The sequencing results showed that the different groups were well clustered (Figure 1A), and the differential proteins were significantly enriched (Figure 1B). As a result, by excluding the protein contained in the medium (Appendix A), 135 were identified in PmCQ2 and 147 in PmCQ6, among which 40 and 49 putative secretory proteins were identified from the supernatant of CQ2 and CQ6, respectively (Figure 1C). The differential expression proteins of high- and low-virulent strains are shown in Appendix A. Moreover, 152 proteins were detected in BHI medium and 99 proteins in Martin medium, with significant differences (Figure 1C); 50 and 32 putative secreted proteins were identified in BHI and Martin media after bioinformatics analysis. Detailed differentially expressed proteins of different media are shown in Appendix A. In summary, we screened a total of 50 putative secreted proteins (Table 1), 22 of which were found in all four groups (PmCQ2-B, PmCQ6-B, PmCQ2-M, PmCQ6-M).

Genome-predicted secreted proteins were consistent with proteome sequencing results. Forty-seven percent of secreted proteins were secreted by the Sec pathway, of which 157 were cleaved by SP I (33%) and 66 were cleaved by SP II (14%). The proportion of Tat pathway secreted proteins was 13%, only 62 proteins. Another 190 were secreted through the non-classical secretion pathway, accounting for 40%. The experimental and genomic prediction results were shown in the figure (Appendix A).

### 3.2. Analysis of Differentially Putative Secretory Proteins between High- and Low-Virulent Strains

The results of cluster analysis showed that the secreted proteins of PmCQ2 and PmCQ6 were slightly different. The secreted proteins of PmCQ2 and PmCQ6 were different in the same medium. In Martin broth, compared with PmCQ6-M, the expression of two proteins in PmCQ2-M were upregulated, namely, BexD (capsular polysaccharide export protein) and IbpA (adenosine monophosphate-protein transferase and cysteine protease), and four proteins were downregulated: htrA (periplasmic serine protease), TonB-dependent hemoglobin transporter, DUF2303 (domain-containing protein), and RP-L23 (50S ribosomal protein L23) (Figure 2A,B). In BHI broth, seven proteins were upregulated in PmCQ2-B/PmCQ6-B, which were dps (starvation-inducible DNA-binding protein), IbpA, LpoA (penicillin-binding protein activator), clpB (ATP-dependent chaperone), prlC (oligopeptidase A), OmpA-OmpF (outer-membrane protein A), and hypothetical protein. Seven other proteins were downregulated in PmCQ2-B/PmCQ6-B, including dppA (ABC transporter substrate-binding protein), PTS-Glc-EIIA (PTS glucose transporter subunit IIA), DUF2303, hypothetical protein, afuA/fbpA (iron ABC transporter substrate-binding protein), TonB-dependent hemoglobin, and MipA/OmpV (MipA/OmpV family protein) (Figure 2C,D). After further analysis, the functions of the above differential proteins are mainly concentrated in iron transport and receptor activity, which may be related to the virulence difference between PmCQ2 and PmCQ6.

### 3.3. Analysis of Differentially Putative Secretory Proteins in Different Media

Previous experiments found that there were differences in virulence and growth levels between PmCQ2 and PmCQ6 cultured in different culture media; therefore, we compared the discrepancy proteins separately. The secreted proteins of *P. multocida* strains in Martin and BHI broth were significantly different, and the screening criteria for differentially expressed proteins were the same as above. The secreted proteins of PmCQ2 and PmCQ6 were differentially expressed in different media. Twelve proteins upregulated in PmCQ2-M/PmCQ2-B were as follows: LpoA, Tuf (elongation factor), DUF2303, hypothetical protein, dppA, OsmY (osmotically-inducible protein), OmpA-OmpF (outer-membrane protein A), phosphoenolpyruvate carboxykinase (ATP), pyk (pyruvate kinase), fumarate hydratase class II, prlC, RP-L1 (50S ribosomal protein L1) (Figure 3A,B). Twenty-two proteins were upregulated in PmCQ6-M/PmCQ6-B, namely, OsmY, trxA (trxM protein), hyPrx5 (hybrid peroxiredoxin), enolase (phosphopyruvate hydratase), rpoA (DNA-directed RNA polymerase subunit alpha), RP-L1, DUF2303, hupA (DNA-binding protein HU-alpha), RP-S1, RP-S5, RP-S7 (30S ribosomal protein), cysK (cysteine synthase A), MipA/OmpV, GBP (major outer-membrane protein), OmpA-OmpF, fumarate hydratase, Tuf, BexD, pyk, slyB (outer-membrane lipoprotein), dppA and hitA; the expression of two proteins were downregulated, namely, RP-L23 and htrA (periplasmic serine protease) (Figure 3C,D). Under more suitable culture conditions (in Martin broth), the dppA, DUF2303, OmpA-OmpF, OsmY, pyk, RP-L1, Tuf (elongation factor Tu) proteins of PmCQ2 and PmCQ6 were both upregulated. In addition, dppA and OsmY have been reported to be associated with oligopeptide uptake of *Streptococcus agalactiae* [14] and immunoreactive virulence factors of *Actinobacillus pleuropneumoniae* [15], respectively. Therefore, we speculate that these proteins may be involved in the virulence regulation of *P. multocida*.

### 3.4. Validation of Secreted Proteins

In comparison with previous transcriptome sequencing results and by reviewing references, a total of 16 proteins were screened as putative protective antigen candidates (Appendix A). Among them, Tuf protein with high expression in vivo was selected for verification. According to bioinformatics analysis, Tuf is localized in the periplasm (Appendix A), with signal peptide and non-classical secretion pathway (Appendix A), and no transmembrane domain (Appendix A), which is consistent with the characteristics of secreted proteins. To further verify the bioinformatics analysis, the expression and subcellular localization of the protein were studied. PCR (Figure 4A) and SDS-PAGE (Figure 4B) detection showed that the Tuf proteins were successfully expressed and purified from *E. coli*. Polyclonal antibody against Tuf was successfully prepared by immunizing mice. According to the immunohistochemical results (Figure 4C,D), rTuf had good antibody specificity and high secretion levels. Immunoelectron microscopy showed that Tuf protein was secreted into the host lung epithelial cells, interstitium, and capillaries after *P. multocida* infection, suggesting that Tuf is a secreted protein of *P. multocida* (Figure 4E,F).

### 3.5. Results of rTuf Protein Biological Characteristics

Bacterial secreted proteins have been gradually discovered and studied as protective antigens. To determine whether Tuf can be used as a candidate subunit vaccine, we then studied its biological characteristics. The purified protein was made into subunit vaccine to immunize mice, and then the serum was isolated by blood collection from the tail vein. The serum antibody titer of mice was determined by the indirect ELISA method. The results showed that the antibody titers of Tuf were 1:51,200, indicating that the screened putative protective protein had good immunogenicity (Figure 5A). After tail intravenous blood collection, mice were challenged with PmCQ2 (1 × 10^7^ CFU) via the intramuscular route. All mice in the control group died, while the mice immunized with rTuf protein still had survivors, and the rTuf vaccine efficacy was 40% (Figure 5B).

Studies have shown that many bacterial proteins induce host cells to secrete related cytokines [16]. For instance, purified rPM0442 protein induces dose-dependent secretion of IL-1β, TNF-α, IL-6, and IL-12p40 [17] in macrophages. According to ELISA detection, the endotoxin-free rTuf protein also has the ability to induce macrophage to secrete cytokines. As shown in Figure 5C–H, rTuf protein induced the secretion of cytokines in a dose-dependent manner within a certain dose range (1~5 ug/mL), including TNF-α, IL-6, IL-17, IL-12p40, IL-1β, and IFN-γ. These results suggest that rTuf protein may be involved in the immune response of macrophages to *P. multocida*, and the specific mechanism needs to be further explored.

## 4. Discussion

*P. multocida* is a Gram-negative opportunistic pathogen, and its serotype A can cause respiratory disease and pneumonia in stressed cattle, resulting in large economic losses to the cattle industry. With the proximity of humans and animals, *P. multocida* may indirectly infect people through animal scratches or bites, leading to diseases such as respiratory inflammation. Therefore, it is of great significance to study the pathogenic mechanisms of *P. multocida*.

Secreted proteins of bacteria play crucial roles in nutrient acquisition, adaptation, intra- and inter-species signaling, and virulence [18]. The pathogenicity of bacteria depends on its virulence factors to a large extent, and some extracellularly secreted proteins are essentially virulence factors, the more classic one is the α-hemolysin of pathogenic *E. coli* [19]. In addition to virulence, bacterial secreted proteins have multiple functions. For example, T2SS promotes the specific transport of folded periplasmic proteins to the outside of the bacteria and targets them by binding to host cells, which helps bacteria adapt to different external environments [9]. Secreted proteins are also useful for adhesion and biofilm formation. Yoshida et al. found that the biofilm-associated protein BapA translocated to the extracellular environment via T1SS in *Paracoccus denitrificans* could change the hydrophobicity of the cell surface and promote adhesion and biofilm formation [20]. Moreover, secreted proteins are involved in the regulation of host immune response. RavD, an effector protein encoded by *Legionella pneumophila*, harbors a deubiquitinase activity specific for linear Ub chains to disrupt host ubiquitination signaling and modulate host immunity [21]. But these secreted proteins have not been reported in *P. multocida*. Previous studies have found that PM0442, an outer-membrane protein, is a virulence factor of PmCQ2, which can regulate the expression of other virulence genes and induce the secretion of inflammatory cytokines in macrophages [17].

We grew *P. multocida* in static cultures to limit cell lysis. Gong et al. found that different cultures would lead to significant differences in fimbrial gene expression, specifically, *Salmonella gallinarum* and *Salmonella enteritidis* expressed the most abundant fimbriae genes in static broth culture conditions [22]. Matthew et al. compared the proteomic data of *Mycobacterium tuberculosis* under static and shaking culture conditions, and found that at least 45 differentially expressed proteins and six proteins were only expressed during static culture [23]. The above studies suggested that static culture might be conducive to more secreted proteins expression. However, there is an apparent problem with the static culture method, which is hypoxia. Although the conical flask was shaken every half hour, the effect of this behavior was unclear. In addition to local hypoxia, changes in pH, available nutrients, accumulation of metabolic waste, and even quorum sensing could affect bacterial protein expression.

The study of secretomes facilitates the discovery of biomolecules. For example, Li et al. used label-free technology to analyze the secreted proteins of rough *Brucella* by comparative proteomics. The OmpW family protein (BAB1_1579) and the uncharacterized protein BAB1_1185 (two differentially secreted proteins) were found to be related to the cytotoxicity of *Brucella* [24]. From two-dimensional electrophoresis, to the later isotope labeling method represented by iTRAQ (isobaric tags for relative and absolute quantitation) and TMT (tandem mass tags), and to the DIA method such as SWATH [24], proteomics analysis has obtained new breakthroughs. Venable et al. conducted a relative quantitative analysis of yeast protein samples, setting a precedent for the application of DIA [25]. The results showed that the amount of protein quantified by DIA was 87% higher than that by DDA (data-dependent acquisition). Compared with the traditional DDA mass spectrometry technology, more low-abundance proteins are obtained, the randomness of acquisition is reduced, as well as extremely high detection reproducibility and stability are achieved. To obtain more comprehensive secreted proteins, we analyzed the protein samples by LC-MS/MS (QE-HFX_DDA mode), and the obtained protein database was used as the database for DIA work, and then performed LC-MS/MS (QE-HFX_DIA mode) analysis for each sample.

Similar to previous research, our research also proved that protein secretion is closely related to the difference of culture medium, and some proteins are secreted in a certain environment. A total of 152 proteins and 99 proteins were detected in BHI and Martin media, respectively. After bioinformatics analysis, 50 and 33 putative secreted proteins were identified. According to the SPI signal peptidase cleavage, 19 proteins were screened, which mainly included virulence proteins, ribosomal proteins, and enzymes related to bacterial growth and metabolism. Fkpa, sialic acid binding protein (SABP), pal and TadD were reported as virulence factors. The N-acetylneuraminate epimerase (nanM) had an N-terminal signal peptide domain and could be secreted out of the membrane. Compared with Martin medium, 10 proteins were upregulated in PmCQ2 cultured in BHI medium. DNA binding protein HU regulates the expression of many genes related to growth, motion, metabolism, and virulence [26]. Phosphoenolpyruvate carboxy kinase (PCK) is an important enzyme in gluconeogenesis that induces amino acid synthesis and energy metabolism. PCK synthesis would increase when the bacterial growth rates were low [27]. From our experimental results, we saw that the growth rate of the two strains in BHI was significantly lower than that in Martin, the expression of PCK in BHI of two strains was higher than in Martin. This may also reflect the importance of PCK. The upregulation of these three factors made us suspect that the environmental pressure of *P. multocida* in BHI medium was high. The nutrients, osmotic pressure, and pH of BHI medium might not be suitable for the growth of *P. multocida*. In addition, the expression of 50S ribosomal protein L1 was increased, because some pathogens would indeed produce many more ribosomal proteins and enzymes to deal with external pressure in response to environmental stress [28]. Compared with Martin medium, BHI medium is relatively poor in nutrition; therefore, bacteria need to express more anti-stress proteins and virulence factors in BHI medium to adapt. These processes require energy, resulting in slower growth and increased virulence in BHI medium.

In-depth research on the pathogenic mechanism of bacteria and the discovery of new virulence factors are carried out for better prevention and controlling of diseases caused by bacterial infections. Vaccines have played irreplaceable roles nowadays, and subunit vaccines have strong immune specificity, single components and clear composition; therefore, they are favored by vaccine developers. We found that PmCQ2 strain had an extracellular lipoprotein (PM0979) with a signal peptide and no transmembrane domain, suggesting that it is likely to be a secreted protein. Ma [29] and Du constructed rPM0979 recombinant protein, with a 60% protective efficacy for PmCQ2, indicating that PM0979 had immunogenicity and was a protective antigen. Some outer-membrane proteins may enter the secretion supernatant of *P. multocida* through a specific secretion mechanism, and play moonlighting roles in pathogenicity and evasion, which provides us with more ideas for exploring the pathogenic mechanism of *Pasteurella*. For instance, previous research on PM0442 found that it could induce macrophages to secrete TNF-α, IL-6, and IL-12 through NF-κB, p38, and ERK1/2 signaling pathways [17]. To obtain more candidate proteins for a *Pasteurella* vaccine, we analyzed the secreted proteins and combined them with the reports on other bacteria, to provide some ideas for screening candidate proteins of subunit vaccines (Appendix A). According to the experimental results, *P. multocida* cultured in BHI secretes more proteins than that in Martin, which indicates a new reference, that when preparing inactivated vaccine, selecting BHI medium to cultivate bacteria may produce more antibodies.

Tuf is the elongation factor Tu (EF-Tu), which is the most abundant bacterial protein and is n-terminally acetylated [30]. EF-Tu is a key component in the translation of most proteins in *Mycobacterium tuberculosis*, which is related to bacterial cell wall synthesis and can be induced in anaerobic and high-iron culture media, closely related to bacterial protein synthesis. At the same time, the abundance and conservation of EF-Tu in bacteria make it recognized as one of the pathogen related molecular patterns by the plant innate immune system [31], and its acetylated N-terminal fragment is used as a recognition element [30]. Therefore, in this study, Tuf protein was prepared into a subunit vaccine to test its protection against *P. multocida* infection in mice. The results showed that the protein had good immunogenicity and stimulated the production of inflammatory factors in mouse peritoneal macrophages, but the specific mechanism of action was not clear and needs to be studied in subsequent experiments.

## 5. Conclusions

In summary, we screened 50 putative secreted proteins through DIA LC-MS/MS combined with bioinformatics analysis. The amount of secreted protein was less correlated with bacterial virulence, but with the bacterial culture medium. *P. multocida* grows better in Martin medium and secretes more protein in BHI to resist external pressure, which provides a theoretical basis for the study of *Pasteurella* pathogenesis. The Tuf selected in this study has excellent immunogenicity and can stimulate the production of inflammatory factors of macrophages, which can be preliminarily determined as the virulence factor of *P. multocida*, but its tangible mechanism needs further study.

## Figures and Tables

**Figure 1 animals-13-03683-f001:**
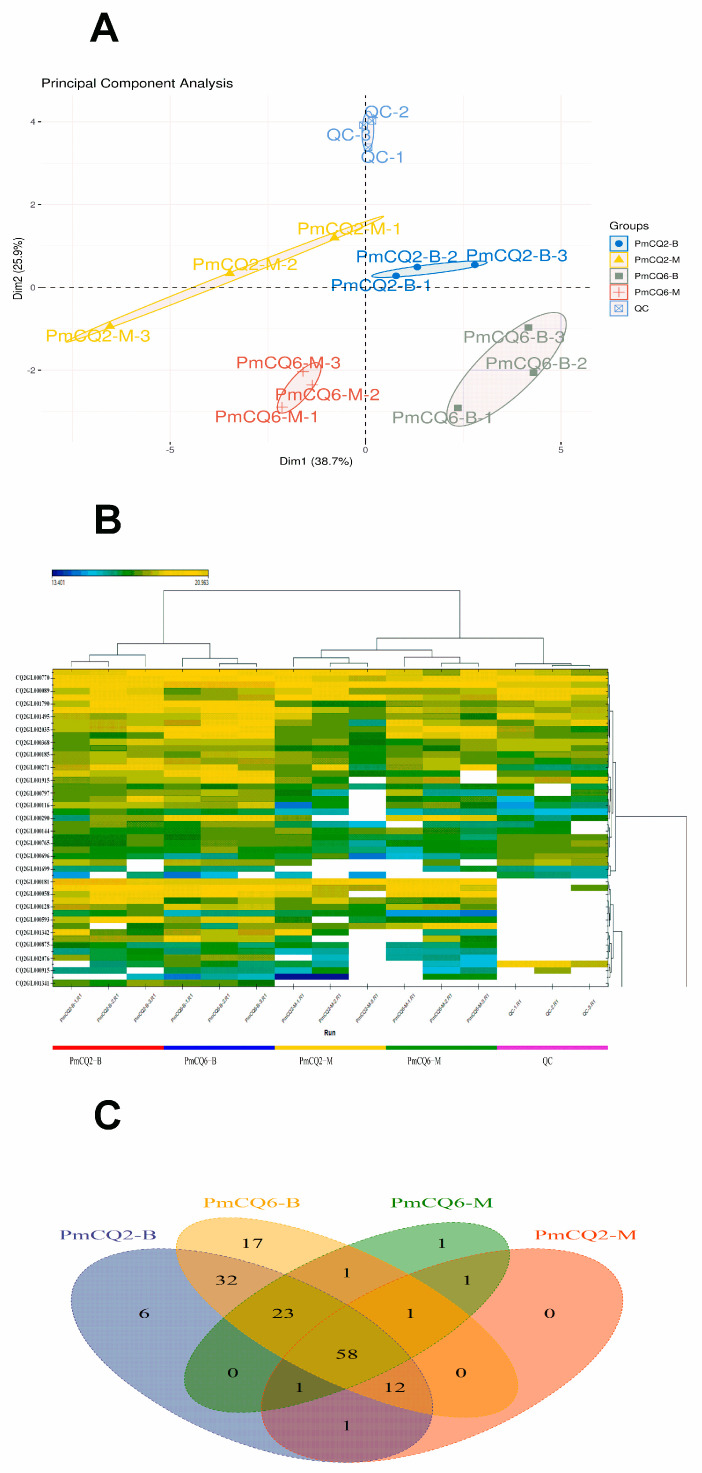
DIA LC-MS/MS secretory proteomics assay results. (**A**) Principal component analysis (Abscissa: the first principal component; Ordinate: second principal component; each point represents a sample). (**B**) Quantitative heat map of protein (Abscissa: sample name; Ordinate: protein login number). (**C**) Venn diagram of differentially expressed proteins between groups.

**Figure 2 animals-13-03683-f002:**
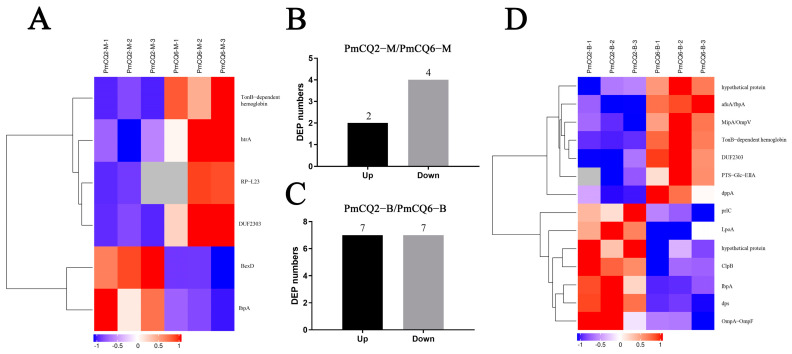
Cluster analysis of differentially expressed proteins between high- and low-virulent strains. (**A**,**B**) Analysis of differentially expressed proteins of high- and low-virulent strains in Martin medium. (**C**,**D**) Analysis of differentially expressed proteins of high- and low-virulent strains in BHI medium. DEP: differentially expressed protein.

**Figure 3 animals-13-03683-f003:**
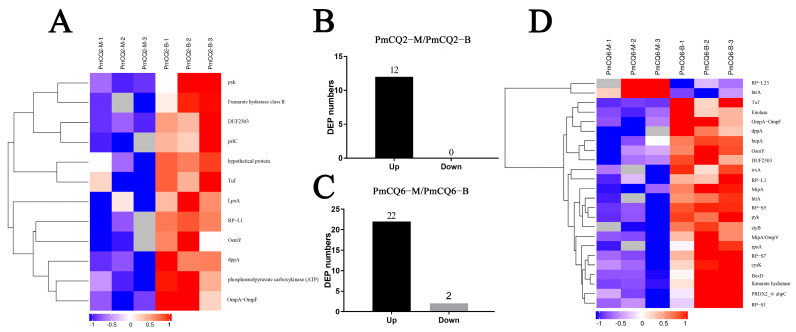
Cluster analysis of differentially expressed proteins in different cultures. (**A**,**B**) Analysis of differentially expressed proteins of PmCQ2 in different media. (**C**,**D**) Analysis of differentially expressed proteins of PmCQ6 in distinct media. DEP: differentially expressed protein. Three replicates were set for each group.

**Figure 4 animals-13-03683-f004:**
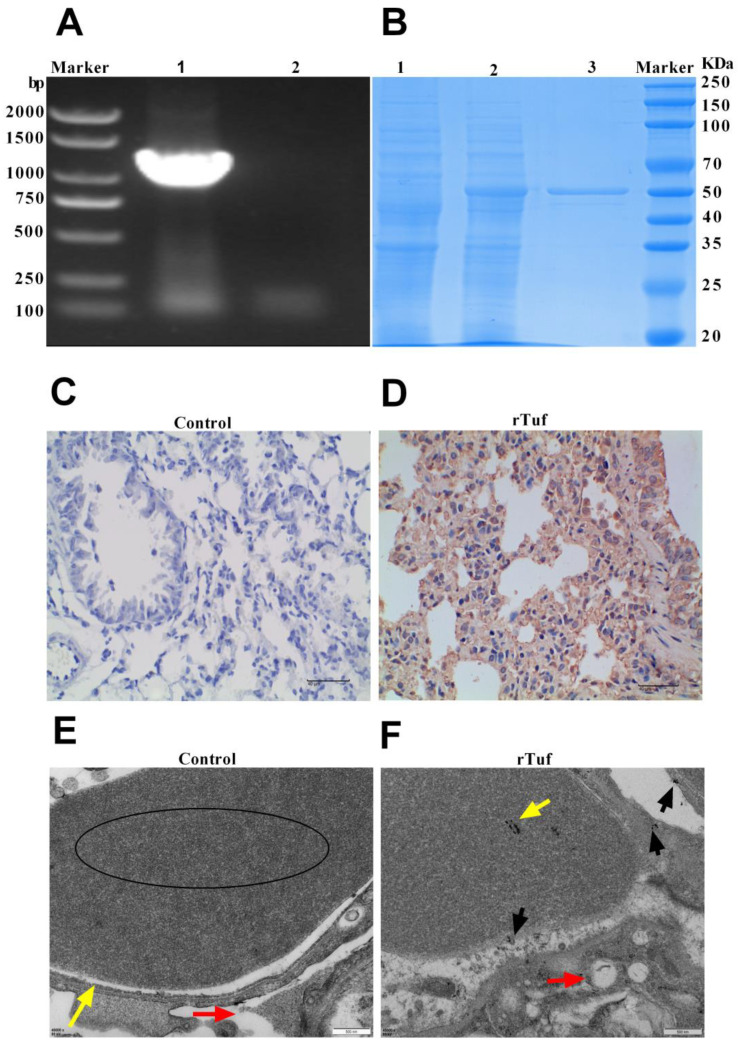
Verification of secreted proteins. (**A**) PCR amplification results of rTuf. Lane 1: Construction of recombinant plasmid rTuf, Lane 2: negative control. (**B**) Results of SDS-PAGE of recombinant protein rTuf. Lane 1: untranslated protein band, Lane 2: expression of the rTuf protein band induced by IPTG, Lane 3: protein band after purification by the HIS-Ni column. (**C**,**D**) Results of immunohistochemical test of mouse lung tissue. The negative cells were blue, the substrate was white, and the positive cells were yellow or brown-yellow. The positive products were mainly distributed in the cytoplasm and intercellular stroma. (**E**,**F**) The localization of secreted proteins was verified by immunoelectron microscopy. (**E**) Capillaries with more normal morphology and structure (↑), mesenchyme (→), erythrocyte (○) ×40,000. (**F**) Capillaries infected by *P.multocida* (↑), mesenchyme (→), gold particles (↑), erythrocyte (○) ×40,000.

**Figure 5 animals-13-03683-f005:**
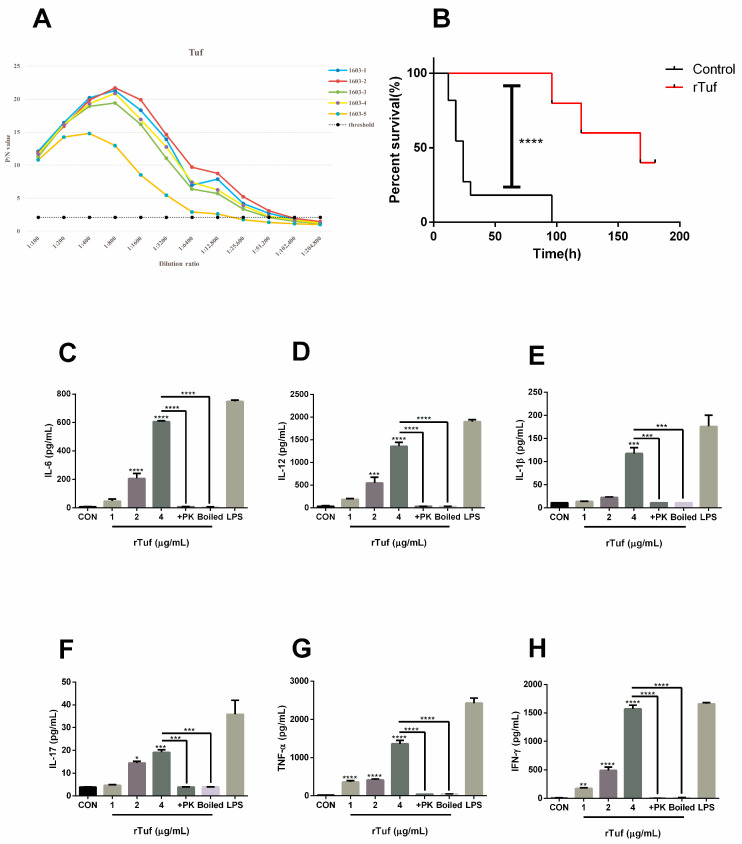
Results of rTuf immunoprotective test. (**A**) Results of antibody titer determination. (**B**) Results of immune protection rate measurement in mice. (**C**–**H**) Secretion level of IL-6, IL-12, IL-1β, IL-17, TNF-α, and IFN-γ in RAW264.7 cells and 1, 2, and 4 μg/mL rTuf; proteinase K (+PK, 50 mg/mL); boiled (15 μg/mL protein); LPS (1 μg/mL) was co-incubated for 24 h. All data were expressed as means ± SEM. * *p* < 0.05, ** *p* < 0.01, *** *p* < 0.001, **** *p* < 0.0001.

**Table 1 animals-13-03683-t001:** Putative secreted proteins.

Protein	Description	Secretory Pathway Prediction	Co-Expressed or Not
CQ2GL000012	50S ribosomal protein L33	Non-classical secretory pathway	Yes
CQ2GL000017	50S ribosomal protein L31	Non-classical secretory pathway	Yes
CQ2GL000368	50S ribosomal protein L20	Non-classical secretory pathway	No
CQ2GL001658	30S ribosomal protein S11	Non-classical secretory pathway	No
CQ2GL001677	50S ribosomal protein L2	Non-classical secretory pathway	No
CQ2GL001678	50S ribosomal protein L23	Non-classical secretory pathway	No
CQ2GL002041	50S ribosomal protein L1	Non-classical secretory pathway	Yes
CQ2GL000089	hypothetical protein	Non-classical secretory pathway	Yes
CQ2GL000090	hypothetical protein	Non-classical secretory pathway	No
CQ2GL000092	hypothetical protein	Non-classical secretory pathway	No
CQ2GL000104	hypothetical protein	Non-classical secretory pathway	Yes
CQ2GL001414	hypothetical protein	Non-classical secretory pathway	No
CQ2GL000675	type I glyceraldehyde-3-phosphate dehydrogenase, GAPDH	Non-classical secretory pathway	No
CQ2GL000697	peptidyl-prolyl cis-trans isomerase, ppiB	Non-classical secretory pathway	No
CQ2GL000950	pyruvate dehydrogenase complex dihydrolipoyllysine-residue acetyltransferase, PdhC	Non-classical secretory pathway	No
CQ2GL001229	peptidase C58	Non-classical secretory pathway	Yes
CQ2GL001642	pirin family protein	Non-classical secretory pathway	No
CQ2GL001744	NAD(P)H-dependent oxidoreductase	Non-classical secretory pathway	Yes
CQ2GL001815	phosphoenolpyruvate carboxykinase, PCK	Non-classical secretory pathway	Yes
CQ2GL002027	glutathione amide-dependent peroxidase	Non-classical secretory pathway	Yes
CQ2GL002051	DNA-binding protein HU, HupA	Non-classical secretory pathway	Yes
CQ2GL000177	Outer-membrane protein A, OmpA	Sec/cleaved by SPI	Yes
CQ2GL000187	nucleotide sugar dehydrogenase, UGDH	Sec/cleaved by SPI	No
CQ2GL000236	htrA protein	Sec/cleaved by SPI	Yes
CQ2GL000270	phosphate acetyltransferase, pta	Sec/cleaved by SPI	No
CQ2GL000380	dipeptide transporter substrate-binding protein, dppA	Sec/cleaved by SPI	Yes
CQ2GL000424	malate dehydrogenase, mdh	Sec/cleaved by SPI	No
CQ2GL000495	C4-dicarboxylate ABC transporter substrate-binding protein	Sec/cleaved by SPI	No
CQ2GL000577	OmpH	Sec/cleaved by SPI	Yes
CQ2GL000589	thiamine ABC transporter substrate binding subunit, tbpA	Sec/cleaved by SPI	No
CQ2GL000696	peptidyl-prolyl cis-trans isomerase (cyclophilin B, ppiB)	Sec/cleaved by SPI	No
CQ2GL000737	translocation protein TolB	Sec/cleaved by SPI	No
CQ2GL000770	MipA/OmpV	Sec/cleaved by SPI	Yes
CQ2GL000846	membrane protein, long-chain fatty acid transport protein, fadL	Sec/cleaved by SPI	No
CQ2GL001241	iron ABC transporter substrate-binding protein, FbpA	Sec/cleaved by SPI	No
CQ2GL001440	RlpA-like protein	Sec/cleaved by SPI	No
CQ2GL002029	FKBP-type peptidyl-prolyl cis-trans isomerase FkpA, fkpA	Sec/cleaved by SPI	No
CQ2GL002076	sialic acid-binding protein, SABP	Sec/cleaved by SPI	No
CQ2GL002078	N-acetylneuraminate epimerase, nanM	Sec/cleaved by SPI	Yes
CQ2GL002179	TonB-dependent hemoglobin/transferrin/lactoferrin family receptor	Sec/cleaved by SPI	Yes
CQ2GL000185	sugar ABC transporter substrate-binding protein	Sec/cleaved by SPII	Yes
CQ2GL000321	osmotically-inducible protein OsmY	Sec/cleaved by SPII	Yes
CQ2GL000324	penicillin-binding protein activator, LpoA	Sec/cleaved by SPII	Yes
CQ2GL000420	glycine zipper 2TM domain-containing protein, slyB	Sec/cleaved by SPII	Yes
CQ2GL000736	peptidoglycan-associated lipoprotein, pal	Sec/cleaved by SPII	No
CQ2GL000898	pilus assembly protein TadD	Sec/cleaved by SPII	No
CQ2GL001790	hypothetical protein	Sec/cleaved by SPII	Yes
CQ2GL000999	DUF882 domain-containing protein	Tat secretory pathway	No
CQ2GL001655	50S ribosomal protein L17	Tat secretory pathway	No
CQ2GL001673	50S ribosomal protein L16	Tat secretory pathway	No

## Data Availability

The raw data have been deposited to iProX—integrated Proteome resources and the accession number is PXD037882.

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
