# Peer review of "Secretome Analysis of High- and Low-Virulent Bovine Pasteurella multocida Cultured in Different Media"

_animals, 2023, doi:10.3390/ani13233683_

Round 1

Reviewer 1 Report

Comments and Suggestions for Authors

The authors highlighted the importance of the secreted bacterial proteins of P. multocida using different culture methods and showed that the bacteria in BHI medium secreted more proteins than in Martin medium. The study used DIA LC-MS/MS and bioinformatics to identify 50 putative secreted proteins. The results provide insights into the pathogenesis of Pasteurella and possible avenues for vaccine development. The manuscript is well written, contains few English errors, and uses scientifically valid methodology - the greatest stregth of this work. The topic is interesting and would be of particular interest to countries with large cattle populations and higher pasturellosis prevalence. The discussion is not rigorous, but I assume that the lack of literature could be the reason for it. Here are some specific comments.

Line 36: Replace the reference with this one "Dabo, S., Taylor, J., & Confer, A. (2007). Pasteurella multocida and bovine respiratory disease. Animal Health Research Reviews, 8(2), 129-150. doi:10.1017/S1466252307001399"

Line 108: replace eating and drinking with feed and water intake

Line 111: replace dead mice with mrtality

Line 113: Which cultures are these? You mean from section 2.1?

Line 166: How do you know that the titer of antierum reached it peak and on which day? Did you perform antibody titration everyday?

Comments on the Quality of English Language

Minor mistakes...

Reviewer 2 Report

Comments and Suggestions for Authors

The manuscript submitted for review presents a study with the aim of exploring the potential virulence factors and protective antigens in secreted proteins in vitro cultured Pasteurella multocida, so as to provide a theoretical basis for vaccine development.

Several studies aimed at identifying the proteins present in the secreted component of a bacterial culture have been published, including one study carried out to detect the secretome of Manhemmya haemolytica (formerly Pasteurella haemolytica). As such, the manuscript presented is not highly original.

See for example:

Ayalew S, Confer AW, Hartson SD, Canaan PJ, Payton M, Couger B. Proteomic and bioinformatic analyses of putative Mannheimia haemolytica secretome by liquid chromatography and tandem mass spectrometry. Vet Microbiol. 2017; 203:73-80. doi: 10.1016/j.vetmic.2017.02.011.

Dwivedi P, Alam SI, Tomar RS. Secretome, surfome and immunome: emerging approaches for the discovery of new vaccine candidates against bacterial infections. World J Microbiol Biotechnol. 2016; 32(9):155. doi: 10.1007/s11274-016-2107-3.

Zubair M, Wang J, Yu Y, Faisal M, Qi M, Shah AU, Feng Z, Shao G, Wang Y, Xiong Q. Proteomics approaches: A review regarding an importance of proteome analyses in understanding the pathogens and diseases. Front Vet Sci. 2022; 9:1079359. doi: 10.3389/fvets.2022.1079359.

The novelty of this study, however, relies on the approach used to gather the data and the targeted bacterial pathogen. By combining the use of data-independent acquisition (DIA) LC-MS/MS with a bioinformatics analysis authors were able to identify a total of 154 proteins obtained from the culture supernatants of two isolates of bovine P. multocida serotype A (high virulent 22 PmCQ2 and low virulent PmCQ6) growth in Martin or BHI media, out of which 50 of them were identified as putative secreted proteins.

The results obtained are thus of interest for the audience of this journal as it was shown that a selected, highly expressed elongation factor Tu (Tuf) protein of P. multocida is also secreted into infected tissues, is highly immunogenic and had moderate protective efficacy against P. multocida infection in a mouse model.

Overall, the manuscript is well written, the laboratory approach concerning the analytical methods utilized is adequate, and sufficient technical details are provided to replicate the work. However, a few items were found by the reviewer in the manuscript so it should be improved before being accepted for publication.

Title

Consider using the following title “Secretome Analysis of High and Low-virulent Bovine Pasteurella multocida Cultured in two Different Media”

Simple summary.

Line 15. It is stated that “Tuf protein was made into a subunit vaccine”. Consider using instead the following Tuf protein was expressed as recombinant antigen and tested as a subunit vaccine”.

 1.     Introduction

Lines 46-48. The statement “transcriptome sequencing analysis of PmCQ2 and PmCQ6 revealed abundant DEGs (differentially expressed genes) [5]” requires the correct reference:

He F, Zhao Z, Wu X, Duan L, Li N, Fang R, Li P, Peng Y. Transcriptomic Analysis of High- and Low-Virulence Bovine Pasteurella multocida in vitro and in vivo. Front Vet Sci. 2021;8:616774. doi: 10.3389/fvets.2021.616774.

This reference should be cited here (and added to the references section). The subsequent references should be re-numbered, including that originally cited as [5] since it is also cited later on in the manuscript (see line 247)

2. Materials and Methods

Line 93. The sentence should start as “A total of 80 C57BL/6 and 50 Kunming mice…” Check if the former type of mice were actually used, as they are not described further in the whole manuscript.

 Line 99. It is indicated that the culture included “medium supplemented with 5 % horse serum,”. Please indicate what kind of procedures were made to delete the horse proteins out of the culture supernatant secretome analysis.

Line 113-114. Please indicate if culture media were supplemented with 5% horse serum

Line 119. Please check if “APPLIED PROTEIN TECHNOLOGY” requires caps.

 Line 122. Speel “HPRP”, first-time use

 Line 130. Same company?. already described above (line 119). does it require an address?

Line 144. Use caps in “Tmhmmv 2.0”. Can use “Finally,” instead of “Definitely”

Line 149. It is mentioned that “Based on bioinformatics prediction and secretion level of the protein,” Tuf was selected for further analysis. Please indicate where the secretion levels (concentrations?) are described and elaborate a bit more on why the Tuf protein was selected for immune protection evaluation in mice. Apparently, it is not differentially expressed in the two different strains of P. multocida??

 Line 161. Correct “using by a toxin removal kit”

Line 172. Spell IHC, first-time use

Line173. Could use “sectioned and mounted in microscopic slides” instead of “sliced”

Line 175. Can use” Slides” or “sections” instead of” slices”. Check the term in lines 177, 178, 182, and 184.

Line 176. Correct the term “The stopped”

Line 181. Use “slides were washed” instead of “washing”. Spell “DAB” substrate, first time use

Line 190. Can use “Grid” instead of “net” in “300mesh nickel net.”

Line 192. Use “nickel grid sections” instead of “nickel mesh slices”

Line 195. Use “and incubated ” instead of “and acted”

Line 198. Use “nickel grids were washed 5 times with PBS” instead of “Then PBS was washed for 5 times”.

 Line 199. Use “as negative stain” instead of “to slightly restrain it”, and “double distilled water was used to wash grids for 5 times”, instead of “and double distilled water was washed for 5 times”

 Line 200. It is stated that “the copper net was collected by USING”. Was it a copper or nickel grid? “USING” does not require caps

 Line 203. Spell “KM”, first-time use. Or is it  KunMing mice?

Lines 205-206. It is stated that “the immune dose of mice varied from 100 μL to 150 μL according to the protein concentration”. Why is it that the volume varied, if only one recombinant protein at 100 µg concentration was used

Line 211 Add “with” in “the mice were challenged… PmCQ2”

Line 215. Spell “hr-P”, first time use

Line 218. Is it a “blocking solution” instead of a “sealing solution”

Line 222. Should it be “anti-mouse” instead of “anti-rat”?

 Line 226. Can use “ELISA reader” instead of “Enzyme-labeled instrument”

Line 229. In mouse macrophage “RAW264.7” specify the source of the cell line, the number of cells used, etc

Line 230. In “Elisa KIT”, specify what type of Kit was used, and if it is commercially available.

3.     Results

Line 248. Use “would be cleaved” instead of “were cleaved”

Line 256 Use “by excluding…” instead of “under the influence of excluding…”

Lines 265-267. The paragraph is confusing. Please specify if out of the 50 secreted proteins, 22 proteins were differentially expressed in the PmCQ2 vs PmCQ6, or in the BHI vs Martin proteomics comparisons.  

Line 272. Legend to Figure 1. In “B: Quantitative heat map of protein (Abscissa: sample name; Ordinate: protein login number).”. Protein login number refers to, for example, the Genbank Accession number?. Characters are too small to be seen in Figure 1B.

 In addition, the authors should explain why is it that Tuf Protein is not included in the differentially expressed proteins described in Table 1, Figure 2, and Figure 3 (or is it that the protein ID label is too small for a reader to be able to see in Figures 2 and 3?

 Line 306. Use “different culture media” instead of “in divergent media”

Line 310. Use “were differentially expressed in different media, instead of “were diversity in different media”

 Line 355- Legend to figure 4. It is stated that The positive products of HSV1 ICP8 were mainly distributed in the cytoplasm”  What does HSV1 ICP8 stand for?

4.     Discussion

Line 396. Can use “extracellularly secreted” instead of “secreted extracellular”

Line 403. Correct species name in “Paracoccus deniticans”

Line 412. It is stated, “Gong et al. found that…”. Reference number? It is not cited in the references section

Line 420. Could use “the conical flask was shaken…” instead of “we shook the conical flask every half hour”

Lines 458, 460, and 476. Use “P. multocida” instead of “Pm”

Line 470. Correct sentence “so they are favored by vaccine developers”.

Line 487. It is stated that Tuf  “is the most abundant bacterial protein… “. Please specify how was this determined.

Lines 503-505. The authors conclude that “The Tuf selected in this study has excellent immunogenicity and can stimulate the production of inflammatory factors of macrophages, but its tangible mechanism needs further study.” Authors should indicate if Tuf protein is considered a virulence factor for P. multocida.

References

Authors should check scientific names in all the titles of the cited articles. They must be typed in italics

Finally. The legend in Supplementary Figure 3. indicates “Tuf bioinformatics analysis. A: CELLO v.2.5 analysis of Tuf protein. B: SignalIP-5.0 analysis of Tuf protein. C: TMHMM analysis of Tuf protein.

 However, in the figure itself, it appears as SeqID  1086005

A quick search performed by the reviewer in the GenBank indicates that this ID refers to “S41525  major ring-forming surface protein precursor - Helicobacter mustelae, later on, replaced by Q48237  Major ring-forming surface antigen precursor.

This protein is a “Cytoplasmic” protein, according to supplementary Figure 3 A.  It is not the Elongation factor Tu, a periplasmic protein, according to the results and discussion section (or is it the same gene product as that of SeqID 1086005?) The authors should clarify this.

Comments on the Quality of English Language

To my opinion, the English language could be improved, but it is possible to understand the manuscript.

Round 2

Reviewer 2 Report

Comments and Suggestions for Authors

The manuscript submitted aims to explore the potential virulence factors and protective antigens present in secreted proteins from in vitro cultured Pasteurella multocida, in order to provide a theoretical basis for vaccine development. The approach used to gather the data and the targeted bacterial pathogen is worth of mentioning. By combining the use of data-independent acquisition (DIA) LC-MS/MS with a bioinformatics analysis authors were able to identify a total of 154 proteins obtained from the culture supernatants of two isolates of bovine P. multocida serotype A (high virulent 22 PmCQ2 and low virulent PmCQ6) growth in Martin or BHI media, out of which 50 of them were identified as putative secreted proteins. The results obtained are thus of interest for the audience of this journal as it was shown that a selected, highly expressed elongation factor Tu (Tuf) protein of P. multocida is also secreted into infected tissues, is highly immunogenic and had moderate protective efficacy against P. multocida infection in a mouse model. Overall, the manuscript is well written, the laboratory approach concerning the analytical methods utilized is adequate, and sufficient technical details are provided to replicate the work.